# Socioeconomic Status and Route Characteristics in Relation to Children’s Exposure to Air Pollution from Road Traffic While Walking to School in Auckland, New Zealand

**DOI:** 10.3390/ijerph18094996

**Published:** 2021-05-08

**Authors:** Mehrdad Rafiepourgatabi, Alistair Woodward, Jennifer A. Salmond, Kim N. Dirks

**Affiliations:** 1School of Population Health, Faculty of Medical and Health Sciences, The University of Auckland, Auckland 1023, New Zealand; a.woodward@auckland.ac.nz; 2School of Environment, Faculty of Science, The University of Auckland, Auckland 1142, New Zealand; j.salmond@auckland.ac.nz; 3Department of Civil and Environmental Engineering, Faculty of Engineering, The University of Auckland, Auckland 1142, New Zealand; k.dirks@auckland.ac.nz

**Keywords:** PNC exposure, children, socioeconomic status, walking

## Abstract

Pedestrian exposure to traffic-related air pollution depends on many factors; including the nature of the traffic on nearby roads and the route characteristics. In this study, we investigated the effects of differences in vehicle fleet and transport infrastructure on children’s exposure to traffic-related air pollution during their morning walk to school in Auckland, New Zealand. Five pairs of routes to schools—each including a low and a high socioeconomic status (SES) school—were selected and traversed over 7–8 days for each pair while measuring particle number concentration (PNC) and GPS coordinates. At the same time, a sample of 200 license plates of cars from each school’s neighborhood was captured using videos, and the age of each car was extracted using an online database. Both the mean age of the cars and the percentage of old cars (>14 years) were found to be higher in the low SES areas. However, neither measure of vehicle age was associated with a significant difference in exposure to PNC for pedestrians. Route features including narrow footpaths and a higher density of traffic lights may be associated with higher levels of PNC exposure. These findings provide insight into the role of urban design in promoting healthy commutes to school.

## 1. Introduction

Walking to and from school can contribute significantly to the physical and mental well-being of children [1]. Previous studies have highlighted the importance of active travel to school in promoting higher physical activity, important for cardiorespiratory and metabolic fitness, stronger muscles and bones, lower levels of body fat, and good mental health [2]. Walking to school can also decrease the risk of adverse psychosocial health and well-being issues as a result of socialization [3].

Road travel, whether by car, bicycle, or by foot, may lead to exposure to air pollution emitted from road vehicles [4], resulting in an increased risk of many life-threatening diseases such as ischemic heart disease, stroke, pulmonary disease, respiratory infections, and lung cancer [5]. In addition to these chronic conditions, acute effects experienced as a result of short-term periods of high levels of pollution on the order of seconds or minutes (spikes) include short-term respiratory discomfort such as wheezing, asthma attacks. or at the extreme, heart attacks and death [6]. Susceptible people (e.g., those with co-morbid heart and lung conditions) may be at higher risk of extreme health effects due to exposure to spikes in air pollution concentrations [4]. Children are also vulnerable to the adverse effects of air pollution due to their high breathing rate relative to body mass, and the immaturity in the lungs and other vital organs [7]. 

Amongst the pollutants of most concern are particle number concentration (PNC), particularly the particles less than 0.1 μm in diameter, which may penetrate deeply into the lungs where they can remain for many years [8]. These particles may enter into the bloodstream, leading to both long- and short-term health consequences, affecting many parts of the body [8]. Thus, many recent studies investigating individual exposure to air pollution from traffic have focused on PNC [4,9,10,11].

Pedestrians typically inhale greater doses of air pollution compared with road users who travel by car for the same journey because of their higher minute ventilation (volume of air breathed per minute) and increased travel time [11,12]. However, pedestrian exposure to air pollution varies considerably, depending on the features of the route taken. For example, the width of the footpath, the nature of the traffic flow on the adjacent road, the presence, distribution, and types of intersections that need to be crossed along the route as well as the nature of the buildings and vegetation present alongside the road in question are all significant contributing factors [13,14]. Specifically, studies have shown that exposures can be reduced by increasing the distance between pedestrians and the road traffic [15], for example, by walking near the building edge rather than along the street edge [16]. Likewise, pedestrians who walk alongside roads are at higher risk of being exposed to air pollution from traffic compared with those who walk along a route separated from traffic, for example, through a park [4]. The time spent walking alongside traffic queueing at intersections also leads to higher levels of exposure compared with other segments of a pedestrian journey [4]. This includes walking close to construction sites where higher exposures to PNC have been observed [9].

The emission rate of vehicles can also vary significantly due to differences in a vehicle’s engine technology, a vehicle’s age, mileage, vehicle model, level of maintenance, and driving behavior [17]. Of these, the age of a vehicle is one of the most important factors, as older vehicles generally emit more pollutants due to accumulated mileage and their older engine technology (emission standards) [17,18,19]. Specifically, the higher accumulation of kilometers travelled results in deterioration of the engine and degradation of emission control equipment in combustion-based vehicles [19]. The gradual improvement in the emission standards of vehicles has meant that newer vehicles produce lower levels of emissions compared with their predecessors [17,18].

It has also been found that there is an inverse correlation between the median age of vehicles and the income of their owners [18]. However, emission rates of vehicles for the same age and model have been found to be higher in areas of high socioeconomic deprivation [17]. This could be due to a number of factors such as poorly manufactured vehicles (e.g., a malfunction in engines) being sold to lower-income people, a lack of money available for vehicle maintenance, vehicles that have a higher mileage for their age (and are thus cheaper to purchase), or perhaps a combination of these [17].

The difference in the pollutant emission rate of vehicles in areas of low and high levels of SES can lead to differential exposure to traffic-related air pollution depending on the socioeconomic deprivation of a neighborhood. These differences may lead to socioeconomic inequities in health, particularly amongst school children walking to school. All other factors being equal, children walking to school located in a neighborhood of low SES can be expected to be exposed to higher concentrations of air pollution than children living in high SES neighborhoods. Furthermore, for children who walk to school every day, both the cumulative exposure to air pollution and the health disparities associated with such differences are likely to be significant. Exposure to spikes (or short-term elevated concentrations) could also lead to acute health effects, independent of the long-term exposure [6]. Consequently, individuals exposed to the same dose of air pollution can be expected to experience different acute health impacts. Thus, children should be guided to choose routes that will help to avoid both spikes in pollution and limit cumulative exposures whilst walking to school [4]. For example, a large reduction in their overall exposure while commuting could be achieved by avoiding interrupted and congested traffic flow such as that created by traffic lights and traffic intersections [11]. 

It has been found that people tend to choose routes associated with a lower level of air pollution exposure when such information is provided to them [20], with travel time and distance being the most important factors associated with route choice [21]. However, the shortest or fastest route, and the route associated with the lowest exposure are not always the same. In an analysis of the factors affecting walkers’ route choices, Bunds et al. (2019) found that pedestrians tend to choose the route associated with the lowest air pollution over the shortest or fastest route. Therefore, providing information about air pollution exposure and route choice to parents, caregivers, and children could be beneficial from the point of view of health.

To date, there has been little research on the impact of a neighborhood’s SES on children’s exposure to traffic-related air pollution while walking to school. Lehn (2017) studied two pairs of schools (with each pair consisted of a low and a high SES school) and found no clear link between the SES of schools and exposure of children to traffic-related air pollution. The size of that study was clearly a limiting factor. Thus, we aimed to fill this research gap by including five pairs of schools. In addition to investigating average exposures, we aimed to determine whether there was a difference in the frequency of spikes for children from high and low SES schools while walking to school, and attempted to identify the potential causes of spikes in pollution.

## 2. Materials and Methods

### 2.1. Location 

The study was undertaken in Auckland, New Zealand, located in a narrow isthmus (−36.848461, 174.763336). Its highest and lowest elevations above sea level are 196 meters and 0 meters, with a mean of 54 meters [22]. Auckland is the biggest city in New Zealand and covers an area of 1102.9 km^2^ (425.8 sq. mi). Based on the 2018 census count, the population of Auckland is 1,534,700 with a population density approximately $1400/{\mathrm{km}}^{2}$ [23]. About one-third of New Zealand’s population live in Auckland [23].

In New Zealand, schools are categorized based on the socioeconomic deprivation of the school’s neighborhood, with the most deprived classified as decile 1 and the least deprived classified as decile 10 [24]. In this study, schools with deciles ranging from one to four were considered to be low decile, while schools with deciles ranging from eight to ten were considered to be high decile. Five pairs of schools were chosen for the study. 

Each pair consisted of one low decile school and one high decile school (Table 1). The distances between the two schools in each pair were less than 2.5 kilometers to ensure similar weather conditions and background levels of pollution. This also enabled the researcher to collect data for the two schools in each pair semi-simultaneously.

Figure 1 presents a map of the locations of the schools chosen for the study. Schools located near significant industrial centers were excluded to avoid influences of other sources of PNC on the measurements. Additionally, schools were chosen that were located more than 1.5 kilometers from the sea as the wind flows tend to be different close to the coast. A walking route for each school trip was selected that was less than 1.5 kilometers in length to ensure it was a realistically walkable distance for school children. Each route consisted of a segment along a major road, typically one of the busiest roads in the schools’ neighborhood as well as a segment along a quieter stretch of road. School pairs were selected so that for each school, the segment along the major road, as much as possible, had the same orientation, and the footpath used was on the same side of the road in order to minimize the impact of differences in wind flow relative to road orientation. All but one pair of schools (Pair 2) were located in one of the central suburbs of the city. 

The routes in each pair were also chosen to have similar conditions along the main segment of the route. However, the number of traffic lights and the width of the footpath were not necessarily the same along each of the routes within each pair (Table 2). For example, for Pair 1, the major segments of the route were similar between schools. Pair 2, the footpath along the route for the high decile school was narrow, and, in some parts, was only on one side of the road. The low decile route of Pair 2 included a wide footpath in the major segment of the route. In Pair 3, the high decile school included a narrow footpath and one set of traffic lights. Based on primary observation, a higher traffic flow was observed on the side of the road opposite the researcher’s path where the vehicles head towards the city center. In contrast, the footpath along the main segment of the low decile route of Pair 3 was wide in the main segment of the route. In Pair 4, there were two traffic light intersections in the low decile route behind which there was typically a long line of queued vehicles, and only one set of traffic lights in the high decile route of Pair 4. In Pair 5, the high decile route included a narrow footpath and three traffic light intersections; the researcher needed to wait at one of these to pass the intersection. However, in the low decile school of Pair 5, there was only one set of traffic lights, and the footpath was wider compared to the high decile route.

### 2.2. Instrumentation

Data collection consisted of the measurement of PNC, the capturing of videos (in order to extract license plates and traffic flow information), and the collection of GPS data while walking to school. A TSI Instrument (P-Trak particle counter 8525) was used to measure PNC, counting particles ranging in size from 0.02 to 1 micrometers [25] and concentrations ranging from 0 up to $5\times 10^{5\;}\frac{\mathrm{pt}}{{\mathrm{cm}}^{3}}\;$ [25]. Data collection was only carried out when the conditions were dry to protect the instrument from damage. The device was set to collect data every second (1 s). During the data collection, the device was held at waist height to approximate the height of the breathing zone of a child. 

Videos of the vehicles driving along the roads were captured using a Galaxy S8+ mobile phone and acquired while walking the routes. The video quality was set to 4 k with 30 frames per second [26] in order to provide sufficient resolution for the car’s number plate to extract the year of manufacture of each car travelling along the routes using the ‘Car Jam’ website [27]. The videos were stored in a micro SD card, and, after data collection, were transferred onto a computer for analysis. The traffic flow rate was measured every day using the captured videos. 

To match the air pollution data to the location at which the measurements were made, a GPS device (Qstarz eXtreme 66-CH series BT-Q1000X) [28] was used. Coordinate data were collected at a one-second interval to match the PNC sampling average. The GPS device was turned on for a few minutes before the start of every traverse to allow the device time to locate the relevant satellites and was turned off a few seconds after the end of the traverse. The coordinate data were exported as an Excel file in order to be matched with the corresponding PNC data points. 

### 2.3. Data Collection

Data collection began at the end of June 2019 and finished at the end of September 2019. School holidays were excluded. The data collection period was from 8:10 a.m. to 9:00 a.m. The clocks of the devices were synchronized before data collection. For each day of data collection, data for one pair of schools were collected. The researcher captured data for the first school (of a pair) at approximately 8:10 a.m. at the start point of the first school, walked the route, and then either walked or took a taxi to the starting point of the route associated with the second school, with travel starting at approximately 8:40 a.m. Traversing each route took between 10 to 20 minutes without swapping between routes. The researcher traversed five pairs of routes over 39 days (a total of seven days for Pair 2 and eight days each for the remainder). Over the data collection period, a total of 83 kilometers was travelled.

### 2.4. Meteorological Data

Weather data were obtained from the nearest meteorological station, obtained from the NIWA (National Institute of Water and Atmospheric Research) climate database [29]. The site was the Motat Ews Station, with the location as indicated in Figure 1. The mean wind speed and temperature were extracted for six ten-minute periods from 8:10 a.m. to 9:00 a.m. for each morning of data collection.

### 2.5. Data Analysis

The captured videos were used to extract the mean age of vehicles and traffic flow. The license plate numbers of 200 vehicles (excluding buses and commercial vehicles) along each of the routes were identified in order to obtain the year of manufacture. The license plates of buses and commercial vehicles were not recorded because these vehicles can be expected to be traversing both low and high SES neighborhoods. The age of the vehicles was then calculated by subtracting 2019 (the year of data collection) from the year of manufacture. The traffic flow on the road adjacent to the footpath was recorded for every journey by video in order to be able to capture license plate numbers. These were then linked to the vehicle fleet characteristics as well as to identify specific events that may have led to observed spikes in PNC. The data were also used to measure the traffic flow rate (vehicles per min) on the main road segment of each of the routes (on the basis of approximately ten minutes of sampling for every observation period). 

For each route, the mean PNC exposure for the whole route was calculated, and the presence of spikes (defined as periods when the particle number concentrations exceeded 100,000 pt/cc [4] in a one-second interval) and the sources of spikes were identified. The mean exposures for the high-traffic part of the routes (the main road) were calculated separately to estimate the emission rates of the vehicles. 

For each day of data collection, the background air pollution during the time of data collection was estimated. The background PNC was measured immediately before and after data collection for five minutes (300 seconds) on a quiet road or in a park (located at least 200 meters away from any heavy traffic) close to the beginning of the first route and near the end of the second route. Given that background PNC is a function of time, a linear fit regression model (background (time)=a×time+b) was used to estimate the background PNC during the period of data collection, assuming a linear trend over time between the start and end of data collection. The 600 data points (2 × 300 seconds background PNC) were used to determine the regression parameters (*a* and *b* in the equation above). The equation was then used to estimate the background PNC for every second of data collection. 

It is expected that a higher amount of air pollution is produced per vehicle in areas of lower SES areas due to older cars and that vehicle age has a significant impact on pedestrian exposure to PNC. In order to determine whether this is the case, exposure data need to be normalized to take into account differences in the traffic flow rates between routes and fleet composition, and adjusted to account for the background PNC, as these can also be expected to impact on pedestrian exposure. If the background air pollution is assumed to be zero, all of the measured PNC exposure is considered to be related directly to the traffic on the road immediately adjacent to where the pedestrian is walking. Otherwise, the background PNC can be subtracted from the exposure measurements. To account for differences in vehicle fleet, it is assumed that HDVs emit on average 47 times the PNC (based on the mean of five studies) of light duty vehicles (LDV’s), under free-flow conditions [30].

On this basis, two different methods of normalization were investigated: normalizing based on traffic flow rate only (NPNC1 = exposure/traffic flow) and normalizing by first adjusting for the background PNC and then by both the vehicle fleet (accounting for the proportion of HDV and LDVs in the fleet) and then the traffic flow rate:(1)$$\mathrm{NPNC}2=\frac{\mathrm{exposure}\;–\;\mathrm{background}}{\mathrm{LDVs}+47\times \mathrm{HDVs}}$$

In this equation, ‘exposure’ is the average PNC along the major road. This is excluding the high exposure from traffic lights and constructions activities, and the high exposures from these places, which were removed from the Excel file using the GPS coordination, the time of data collection, and the videos. The ‘background’ is the background PNC estimate at the time of data collection (as described above), and LDVs and HDVs represent the traffic flow rate of heavy duty vehicles and light duty vehicles, respectively, as observed from the video capture at the time of data collection.

Paired t-tests were used to compare the two groups of routes (those associated with high decile vs. low decile schools) regarding the mean exposure associated with the whole route, the number of spikes, and NPNC1 and NPNC2. All data were saved in Excel format to simplify further analysis. MATLAB software was used for all statistical analyses. ArcGIS 10.7.1 [31] was exploited to generate the maps.

## 3. Results

### 3.1. Meteorological Parameters

Ten-minute averaged temperature and wind speed data were retrieved from the NIWA database for the period from 8:10 a.m. to 9:00 a.m. each day from 27 June to 26 September 2019 [29]. The time series of the daily mean (from 8:10 a.m. to 9:00 a.m.) temperature and wind speed can be seen in Figure 2. 

The time series included 91 data points (consisted of the sampling and non-sampling days). The temperature and wind speed were observed to be highly variable between days. However, the trend was that of an increasing temperature over time, which is consistent with the progression of seasons from winter to spring for locations in the Southern Hemisphere.

The bar plot of wind speed and temperature for each pair of routes can be seen from Figure 3. As expected, there was no significant different in either temperature (t = 0.29, *p* = 0.78) or wind speed (t = 0.29, *p* = 0.79) between low and high decile schools. 

### 3.2. The Age of Vehicles

A bar graph of the mean age of each set of 200 cars, along with the standard error of the mean, is presented in Figure 4. 

The average age of vehicles associated with each school neighborhood ranged from 9.5 to 12.5 years. The age of the cars for each high decile school was compared to its paired low decile school. A paired t-test showed that there was a significant difference between the mean age of cars for the low and high decile schools (t = −5.60, *p* = 0.005), indicating that the mean age of cars in the neighborhood of low decile schools was significantly higher than the mean age of vehicles in their paired high decile schools. 

For the purpose of further analysis, cars older than 14 years old (manufactured before 2005) were assumed to be ‘old, as the mean age of vehicles in New Zealand is approximately 14 years [32]. On this basis, the percentage of old cars observed along the road of each route was also analyzed as an alternative measure, and is illustrated in Figure 5. 

The percentage of old cars associated with the low decile schools was found to be higher than those associated with the high decile schools based on a paired t-test (t = −4.38, *p* = 0.01). Thus, the vehicle fleets were both older and consisted of a higher percentage of old vehicles in the neighborhoods of low decile schools compared with those of the high decile. 

### 3.3. Traffic Flow 

The traffic flow on the road adjacent to the footpath was recorded for every journey by video in order to be able to capture license plate numbers. These were then linked to the vehicle fleet characteristics as well as to identify specific events that may have led to observed spikes in PNC. The data were also used to measure the traffic flow rate (vehicles per min) on the main road segment of each of the routes (for approximately ten minutes). The bar plots associated with the traffic flow of all vehicles and HDVs are presented in Figure 6. 

No significant difference was found between low and high decile schools regarding traffic flow rates (t = 1.96, *p* = 0.12) and the traffic flow rates of HDVs specifically (t = −0.22, *p* = 0.83) when comparing the mean for each route across all five pairs of schools.

### 3.4. PNC Measurement

In this study, the hypothesis that older vehicle fleets resulted in higher air pollution exposures was tested by measuring and comparing PNC along the walking routes for each of the pairs of schools. 

Figure 7 presents a boxplot of the log-transformed PNC exposure for the whole route. The results suggest that there was significant variability in measured PNC from one day to another for a given route. The lowest and highest daily average PNC were 1900 ± 3600 pt/cc (the high decile route of Pair 1 on Day 1) and 57,700 ± 24,100 pt/cc (the low decile route of Pair 4 on Day 1). 

The bar plot of mean PNC exposure for each route can be seen from Figure 8. The average of the mean PNC for each school was 17,800 ± 11,900 pt/cc. The highest exposure route (27,800 ± 17,700 pt/cc) was observed in the low decile route of Pair 4 where two traffic lights interrupted the traffic flow, the footpath was narrow, and a high number of HDVs existed. On the basis of a t-test, no significant difference was found in the mean exposure between low decile and high decile routes (t = 0.03, *p* = 0.98). On the basis of these findings (the significant differences in the age of cars and no significant difference in traffic flow and exposure to PNC between low and high decile routes), the age of the vehicle fleet does not have a significant impact on the air pollution levels to which people are exposed when in the road corridor, over and above the traffic flow rate. On the basis of t-tests performed separately for each pair of schools, no significant difference was found in the exposure to air pollution between low and high decile routes. The one exception was in the case of Pair 4 (t-test, P-value <0.01), which can be attributed to the mentioned characteristics (narrow footpath and the traffic lights) of the low decile route of Pair 4. The lowest exposure route (13,500 ± 6,400 pt/cc) was recorded in the low decile route of Pair 3 where the footpath was wide and the distance between traffic and the farthest edge of footpath was more than six meters. The last segment of the lowest exposure route (one third of the whole of the route) consisted of a quiet street. 

### 3.5. Number of Vehicles and Normalized PNC

The mean NPNC1 for each school can be seen in Figure 9. A paired t-test comparing the normalized PNC between high and low decile schools showed no significant difference (t = 0.32, *p* = 0.76). The highest NPNC1 was found in the low decile route of Pair 4. On this route, footpaths were narrow, and two traffic lights caused vehicle congestion in the route. We also calculated NPNC2 by considering the background PNC as well as the higher emissions rate from HDVs. 

The calculated NPNC2 for each route can be seen in Figure 10. There was found to be no significant difference between the high and low decile routes (t = -0.86, *p* = 0.44). The highest normalized PNC was observed in the high decile route in Pair 2, followed by the low decile route in Pair 4 and the low decile route in Pair 2.

### 3.6. Spikes

In order to identify spikes in the time series, the time at which PNC exceeded 100,000 pt/cc was noted. If two such spikes occurred over a period of less than 10 seconds, they were considered to be the same event, otherwise they were considered to be two separate events. Based on the timing of the events, the video recording of the traffic was used to identify the possible cause of the spike. Figure 11 shows an example of finding the spikes from the time series. 

In total, 94 spikes were observed over the 78 periods of data collection (39 days × 2 routes). As shown in Figure 12, the most significant contributor to spikes was heavy commercial vehicles such as buses and trucks, which accounted for 25.5% of the spikes. The second most frequent cause of spikes (24.5%) was a wave of vehicles passing, for example a queue of vehicles accelerating when a traffic signal turned green. Construction activities, including emissions from vehicles on construction sites and the use of construction equipment nearby, was also found to be another significant contributor. Light commercial vehicles (LCVs) were also found to be a contributor to spikes in PNC (7.4%). 

Of the 94 spikes, 50 were associated with high decile routes (53% of spikes) and 44 were associated with low decile routes (47% of spikes) (Figure 13). There was found to be no significant difference between low and high decile routes regarding the number of spikes (t = −0.40, *p* = 0.71). The maximum number of spikes was observed along the route of the low decile school of Pair 4 (17 spikes), which was also the highest exposure route. The lowest number of spikes was observed in the low decile route of Pair 5 (three spikes), followed by the low decile route of Pair 3 (four spikes). In both of the routes, the footpaths were wide, and there was significant separation between the road and the footpath for most of the routes that enabled the walker to avoid exposure to pollutants emitted from passing vehicles. 

## 4. Discussion

This study set out to assess the impact of a neighborhood’s SES on children’s exposure to PNC whilst walking to school. On the basis of measurements made at five pairs of schools, it was found that, on average, schools in low SES neighborhoods were exposed to older vehicle fleets on the roads alongside the walking routes relative to high SES neighborhoods in the same vicinity. However, this was neither a significant predictor of children’s mean exposure to PNC—even after adjusting for differences in the traffic flow rate —nor for the number of spikes in air pollution experienced. Of all the schools investigated, the highest and third highest NPNC2 levels were associated with high and low neighborhood SES schools in the west of Auckland, respectively. Among the low decile schools, the average age of cars in West Auckland was the highest, and likewise for its high decile pair when compared with the other high decile schools monitored. Therefore, the age of the fleet may observably affect exposure to PNC.

However, the results suggest that other factors beyond the age of vehicle fleet are more significant contributors to differences in exposure along walking routes. These include route characteristics such as traffic lights that interrupt the traffic flow [4,11,33], differences in the proportion of HDVs [4], the width of the footpath [15,34], and the presence of barriers on the sides of the road [16,35].

Waves of vehicles and the passing HDVs were the main contributors to the highest number of spikes and high exposure to PNC. The highest mean exposure route and the highest NPNC1 were on a walking route along a highly-congested road with vehicles queuing at traffic lights and a high number of HDVs. Since HDVs generally burn diesel, they emit higher levels of PNC, resulting in higher levels of PNC [36]. However, LCVs may also run on diesel fuel and may contribute to spikes in PNC. This is consistent with the findings of Dirks et al., who found that passing trucks and buses were the main source of spikes in pedestrian exposure to PNC [4]. The intersections along the routes resulted in congestion at traffic lights, causing vehicles to move in waves. The interruption of traffic flow such as the queuing of vehicles for intersections has also been identified in other studies as a cause of high exposure and spikes in PNC [4,11,33]. Thus, routes with no traffic lights and roads with a lower number of HDVs should be considered when planning walking routes.

Furthermore, the widths of footpaths and roads can also play an important role in pedestrian exposure to traffic-related air pollution. Walking routes with wider footpaths separating pedestrian and road traffic were associated with the lowest levels of PNC exposure and lowest numbers of PNC spikes. On the other hand, walking routes with narrow footpaths and roads increased the proximity between the pedestrians and the source of pollution (traffic), resulting in higher exposure. A direct relationship between distance from traffic and exposure to air pollution has been identified in previous studies [15,34]. In addition, narrow roads also slowed down the traffic, increasing the possibility of congestion and lower levels of vehicle-generated turbulence for dispersing pollutants. 

The presence of barriers flanked on sides of the roads (street canyon) can impede dispersion of air pollution, resulting in an increase in exposure to air pollution for pedestrians. The route with the highest normalized PNC (NPNC2) consisted of a wall on one side of the road and vegetation coverage on the other, impeding the dispersion of pollutants within the street canyon and increasing the possibility of pollutant accumulation occurring. Studies by Kaur et al. and Ai et al. both mention pollution accumulation in street canyons and the increase in exposure to air pollution associated with walkers [16,35].

Construction activity is also considered to be a significant source of PNC [37]. Construction sites were the reason for approximately 12% of the spikes in PNC observed in this study. These spikes can be attributed to demolition work, cement loading, or HDVs such as trucks and crawl loaders. Other recent studies have also identified construction sites as a source of spikes in PNC exposure [9]. Thus, it can be expected that avoiding construction sites could help to reduce overall exposure to PNC in walking journeys.

A statistically significant difference in exposure to PNC between low and high decile routes (over the whole of the route) was only observed in one of the pairs (the average difference was 10600 ± 8400 pt/cc). The effect of schools’ SES (two pairs of schools) in walkers’ exposure travelling to school was also investigated by Lehn (2017). The results showed no trend in exposure to PNC between low and high decile routes; the low decile route showed higher exposure in one pair (the differences were 14,100 ± 4000 pt/cc), while in the other pair, an opposite trend was observed (the difference was −27,800 ± 13,400). However, in this study, the opposite trend in exposure to PNC in one of the pairs was attributed to the characteristic of the route (the congestion of vehicles in high decile route), a feature that is consistent with the results of our study. The mean of all of the measurements made in our study (17,800 ± 11,900 pt/cc) was similar to the results of the study conducted by Dirks et al. (2018) in Auckland, who reported the mean level of 16,500 ± 5200 pt/cc. However, Lehn (2017) measured a mean of 32,900 ± 13,300 pt/cc. Therefore, despite the consistency of the city, weather conditions, and timing of measurements, PNC exposure can discernibly vary between two routes in the same neighborhood. Therefore, the built environment and the vehicle fleet both have a crucial role to play in exposure to air pollution for pedestrians. 

There are a number of important limitations of this study. First, the data from each of the schools in the pairs were not collected simultaneously, but one immediately after the other, as there was only one person involved in the data collection. Second, the data collection occurred over a period of only 7–8 days for each school pair. As a result, the range of possible weather conditions was not diverse and limited to the winter months when air pollution levels tend to be higher in Auckland. However, this is comparable to other similar studies, with one measuring data for five days [11] and another for nine days [4]. Third, while using five pairs enabled us to compare low and high decile schools, they were mostly located in central Auckland, and not necessarily representative of the entire Auckland region. Therefore, in future investigations, it would be helpful to compare PNC exposure in other areas where sharper contrasts in the mean age of vehicles are found. Finally, PNC was measured only in one direction to school and not the return journey in the afternoon. 

Emissions from vehicles in New Zealand are relatively high compared to other developed countries such as the United Kingdom (UK) [38]. A study conducted in Bradford, UK and Auckland, New Zealand suggested that by adopting emissions legislation similar to that of the UK, New Zealand could potentially decrease PNC by a factor of 10 [38]. Considering Auckland’s favorable weather conditions (windy weather), this could significantly improve the overall quality of air in Auckland. However, it is important to consider the relative age of vehicles between the two countries. The average age of passenger vehicles in New Zealand is about 14.1 years [32], while the average age of light passenger vehicles in the UK is 9.1 years [39]. Therefore, one would expect that at least some of the difference in exposure observed between pedestrians in New Zealand and in the UK could be attributed to the age of the vehicle fleet.

The findings of this study have a number of important implications for enabling healthier walks to school through air pollution exposure minimization. Some of these suggestions include the construction of wider footpaths for pedestrians, the introduction of barriers between walkers and road traffic, and other transport infrastructure to increase the separation between walking routes and congested roads. Ideally, routes from home to schools should avoid heavy flows of traffic, particularly HDVs, and avoid intersections and construction sites.

## 5. Conclusions

This research contributes to an understanding of the effect of neighborhood SES and route characteristics on children’s exposure to PNC whilst walking to school. From this study, it was found that, despite the higher percentage of old cars observed along routes associated with low SES schools, there were no significant differences in the frequency of spikes in PNC nor in the mean PNC between routes associated with low and high SES schools, even after adjusting for differences in the traffic flow rate along the adjacent road. The results of this study suggest that the characteristics of routes such as road and footpath width and the number of intersections in the road have a more significant role to play in pedestrian exposure than the average vehicle age. Air pollution levels along narrow footpaths and roads may be associated with higher pedestrian exposures to PNC, with a wider footpath beneficial due to the increased separation between vehicles and pedestrians. Avoiding roads with high numbers of HDVs and intersections would also be helpful to minimize the likelihood of exposure to spikes and reduce the mean exposure to PNC across the whole route. 

There are many health benefits, both mental and physical, associated with walking to school. Where possible, the government should further support walk-to-school initiatives by providing an adequate infrastructure to do so where possible including separate routes for walking to decrease children’s exposure. Additionally, parents and children should be supported in the selection of safe routes to school, taking into account the risk of exposure to air pollution. 

## Figures and Tables

**Figure 1 ijerph-18-04996-f001:**
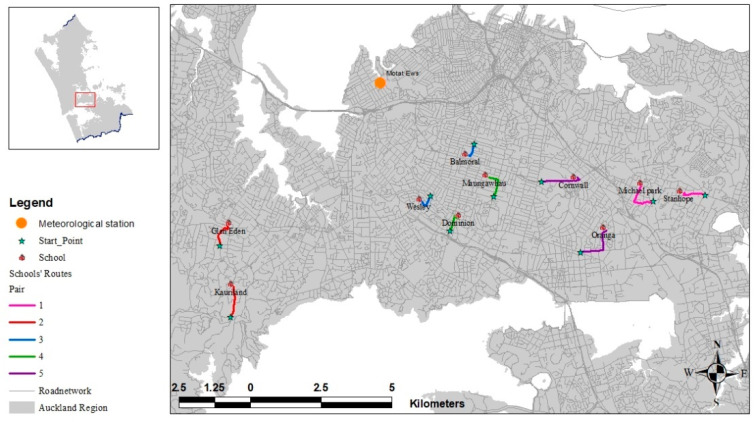
Map of the routes for Pairs 1 to 5. The green stars represent the starting points of each route. Schools are represented as red houses. Orange dot represents the meteorological station.

**Figure 2 ijerph-18-04996-f002:**
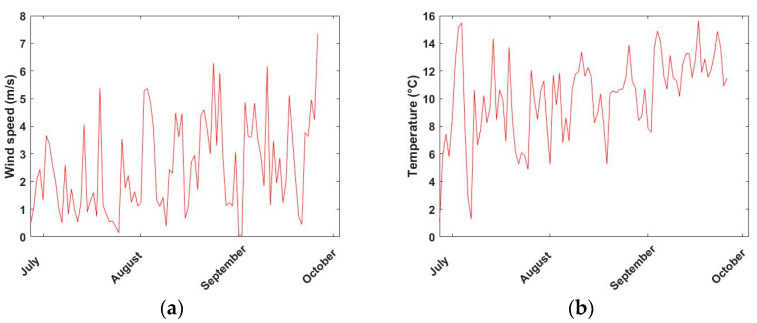
Time series of temperature and wind speed (each average of data between 8:00 to 9:00) over the three-month period of data collection in Motat Ews Station (Retrieved from NIWA [29]) (**a**) wind speed (**b**) temperature.

**Figure 3 ijerph-18-04996-f003:**
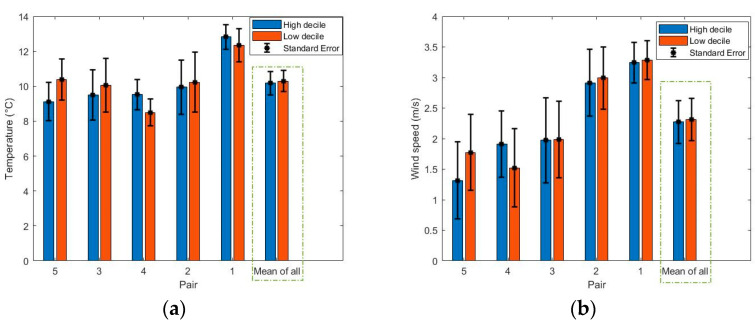
Bar plot of the (**a**) temperature and (**b**) wind speed for each of the pairs of schools around the time of data collection.

**Figure 4 ijerph-18-04996-f004:**
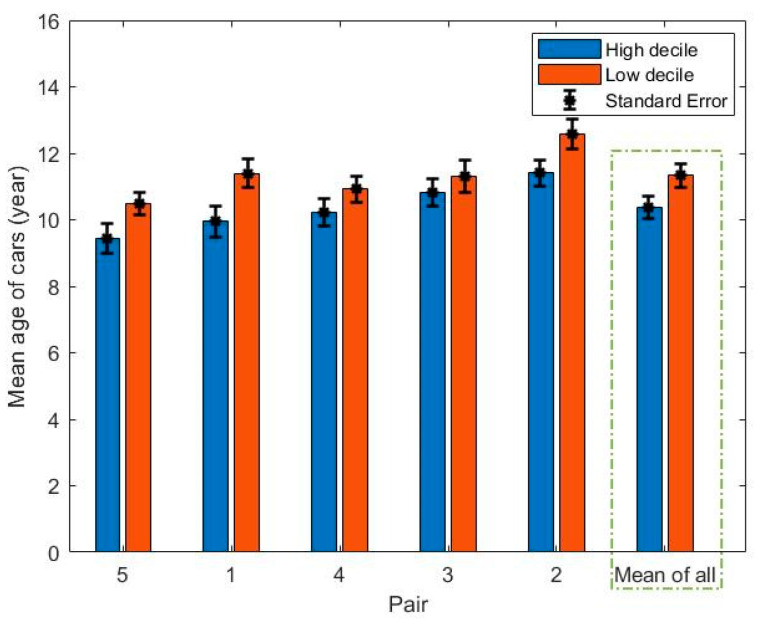
Bar plot displaying the mean age of cars in years for the observed cars in the neighborhood of each school. The error bars are the standard error of the means.

**Figure 5 ijerph-18-04996-f005:**
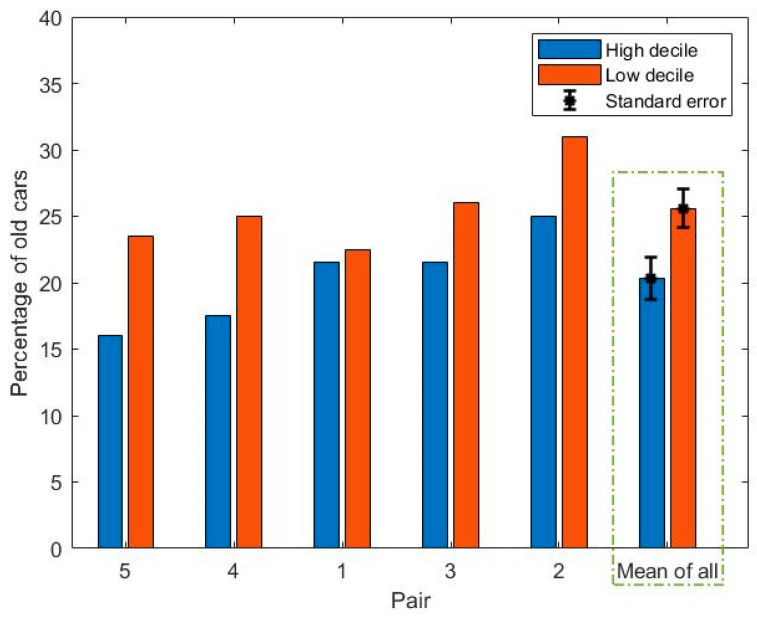
Bar plot of the percentage of old cars (manufactured before 2005) for the roads along each of the school routes.

**Figure 6 ijerph-18-04996-f006:**
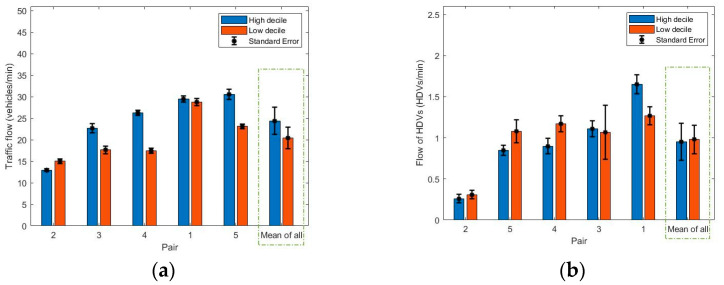
Bar plot of (**a**) traffic flow of all vehicles (vehicles/min), (**b**) traffic flow of HDVs (HDVs/min) along the main road segment (high traffic component) of each route collected over the 7–8 days of observation.

**Figure 7 ijerph-18-04996-f007:**
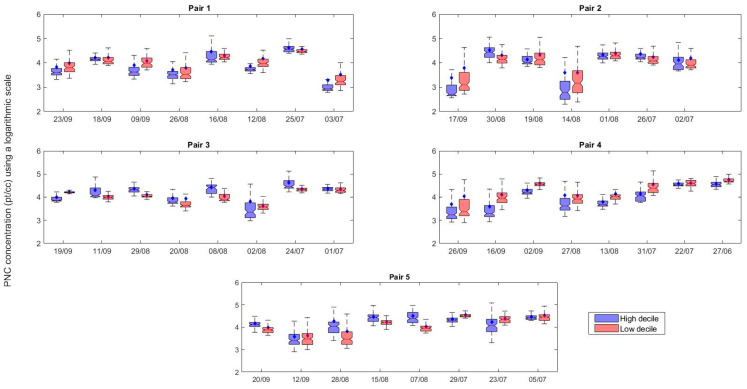
Boxplot of log transformation of measured PNC along the whole routes for each pair of schools. The blue boxes indicate the high decile schools and red boxes indicate the low decile schools.

**Figure 8 ijerph-18-04996-f008:**
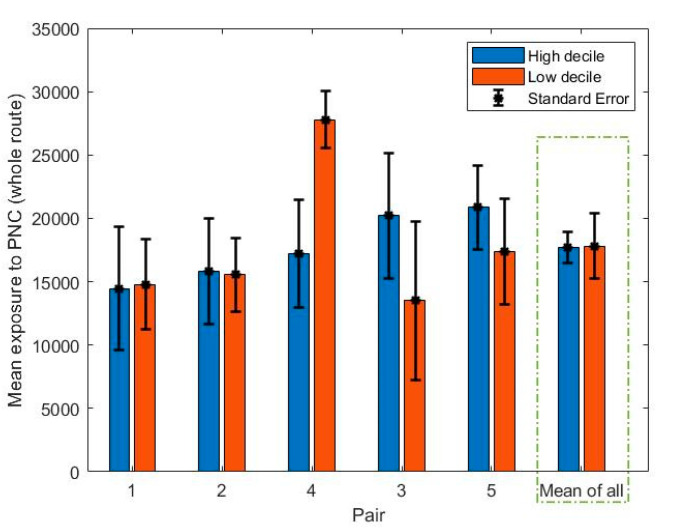
Bar plot of mean exposure to PNC for each pair.

**Figure 9 ijerph-18-04996-f009:**
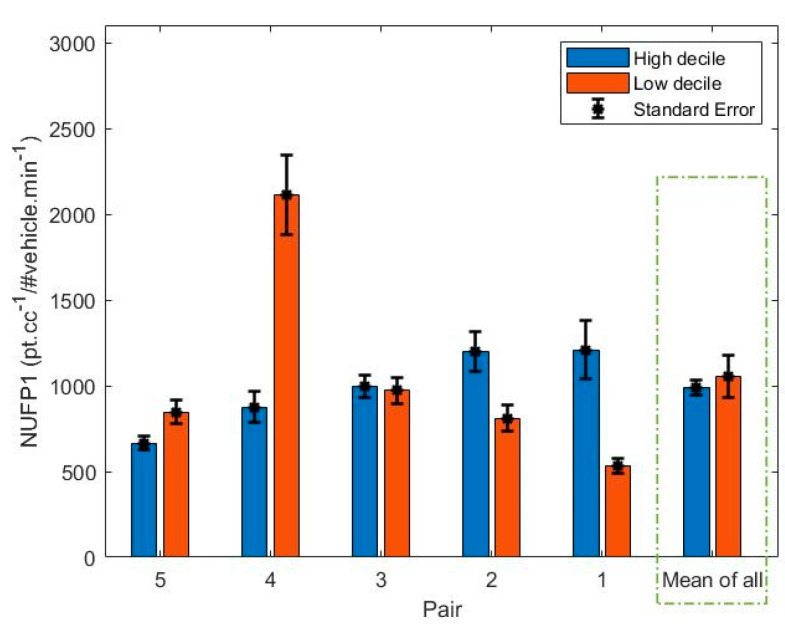
Bar plot of NPNC1 for each pair.

**Figure 10 ijerph-18-04996-f010:**
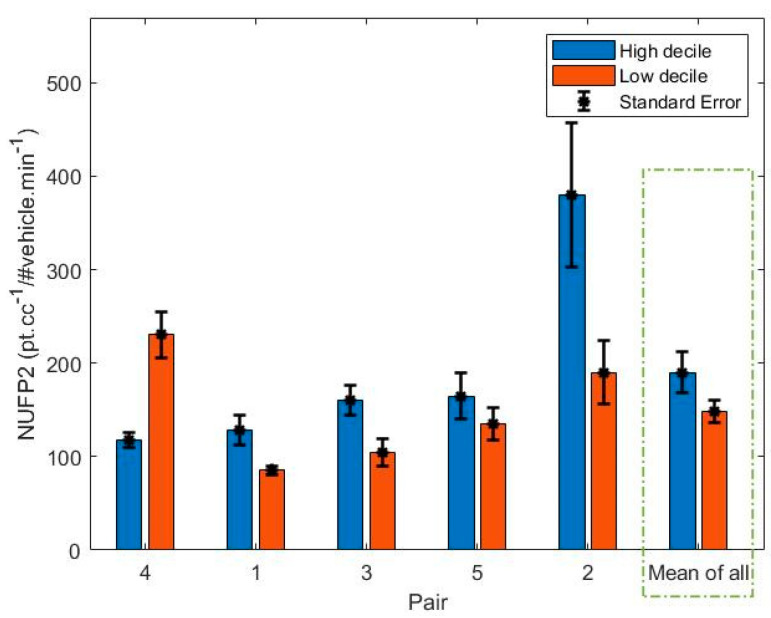
Bar plot of NPNC2 for each pair.

**Figure 11 ijerph-18-04996-f011:**
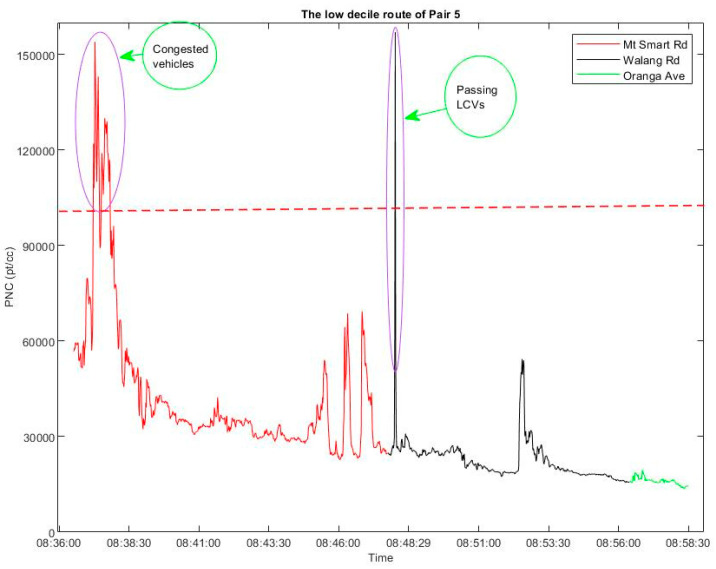
Sources of PNC spikes on the low decile route of Pair 5 (using the data collected on 5 July). The red line represents the measured PNC in the main segment of the route. The black and green lines are the data collected through quiet streets.

**Figure 12 ijerph-18-04996-f012:**
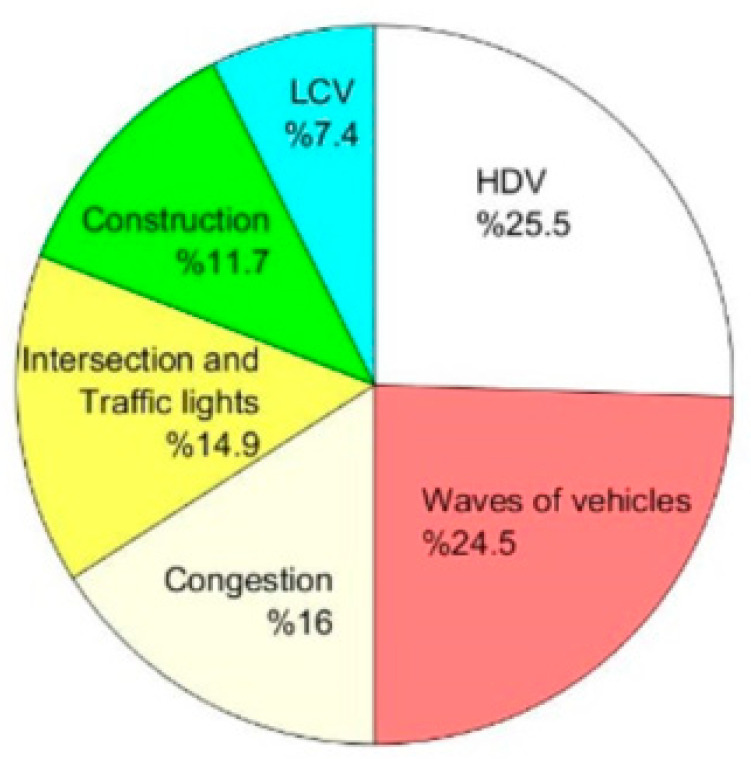
The percentage of each contributor to the spikes.

**Figure 13 ijerph-18-04996-f013:**
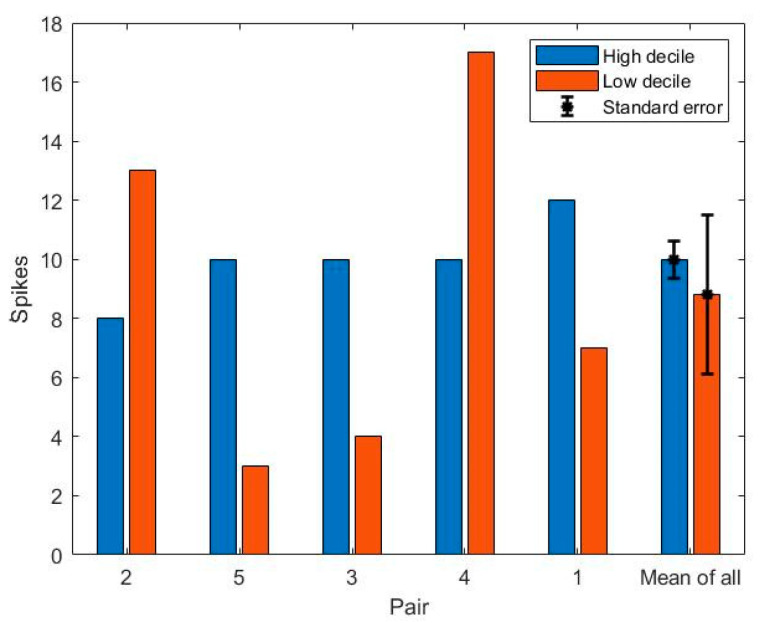
The number of spikes in PNC observed for each pair of routes.

**Table 1 ijerph-18-04996-t001:** The selected pairs of primary schools, their decile, the distances of the school routes for each school, and the distance between the schools in each pair.

Pair Number	High Decile School–Decile–Distance of Routes (km)	Low Decile School–Decile–Distance of Routes (km)	Distance between Two Schools (km)
1	Michael Park School–8–1.42	Stanhope Road School–4–0.87	1.4
2	Kaurilands School–8–1.21	Glen Eden–2–1.1	1.9
3	Balmoral School–9–0.69	Wesley Primary–1–0.68	2.3
4	Maungawhau School–10–0.97	Dominion Road School–3–0.70	1.7
5	Cornwall Park District–9–1.44	Oranga School–3–1.50	1.9

**Table 2 ijerph-18-04996-t002:** The characteristics of the routes (the width of footpath and road are measured in the main segment).

	Low Decile			High Decile		
Pair Number	Width of Road (m)	Width of Footpath (m)	Number of Traffic Lights	Width of Road (m)	Width of Footpath (m)	Number of Traffic Lights
**1**	15	1–2	1	13	1–3.5	1
**2**	6.5–12.5	3.5–5.5	1	10–14	1.5–2	0
**3**	13	5–5.5	0	13.5	3–3.5	1
**4**	12.5	3	2	13.5	3.5–4	1
**5**	13.5	3.5–4	1	17.5	2.5–3	3

## Data Availability

Data are available through a Appendix A.

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
