# Peer review of "Socioeconomic Status and Route Characteristics in Relation to Children’s Exposure to Air Pollution from Road Traffic While Walking to School in Auckland, New Zealand"

_ijerph, 2021, doi:10.3390/ijerph18094996_

Round 1

Reviewer 1 Report

Thank you fot this paper. The article analyzes the impact of socioeconomic status and route characteristics on children's exposure to air pollution from road traffic on the way to school. It deals with issues important from the point of view of children's health, and at the same time the problem of spatial planning.

The subject is difficult, because the current paths leading to schools cannot be changed, and spatial planning often has different priorities. But it is important for local authorities to know what really negatively affects the quality of the air that residents, especially children, breathe.

The presented analyzes and results are clear, and the conclusions clearly show that the biggest problem is the roads to schools located next to busy streets, especially with many crossings and traffic lights.

The lack of influence of the age of cars seems to be interesting, but it might be worthwhile to conduct research on a larger scale in the future, in places where old cars are at least 20 years old. Then these differences would most likely be visible.

Author Response

We appreciate the careful review and constructive suggestions. It is our belief that the manuscript is substantially improved after making the suggested edits. Please find below the response to your concerns. We hope that you find our responses satisfactory and that the manuscript is now acceptable for publication. All line numbers refer to the revised manuscript file with highlighted changes.

The lack of influence of the age of cars seems to be interesting, but it might be worthwhile to conduct research on a larger scale in the future, in places where old cars are at least 20 years old. Then these differences would most likely be visible.

Thank you for the suggestion. We agree with this comment. Therefore, we added some lines to emphasize this point. Please see lines 450-451: “Therefore, in future investigations, it would be helpful to compare PNC exposures in other across where sharper contrasts in the mean age of vehicles are found”.

Reviewer 2 Report

Introduction - The introduction is too long with some irrrelevant redundancies. Paragraphs after line 67 can be cut in half.

Material and Methods - The authors give a detail description related to the traffic design assessment , however they presente a row description in relation to the UFP measurements. For exemple they never mention the distance from the ground to the  the UFP equipment inlet and how the equipement is transport along the road walk, as well the TSI particle model used TSI-P Track 8525) and operation cautions during transport in order to avoid interferences. The mode of log data acquisition depend the objectivs of measuement - source detection (every second), sample mode is recomended at least every 10 seconds (average measurement). 

Data Analysis - the methodology described for the processing of UFP data is confusing and difficult to establish parallelism with the data presented in the Results chapter. They apply assumptions not tested and evaluated in relation to the observed concentrations ​​(example - background concentrations - sensitivity analisys). 

Results - a weak treatment of the UFP concentrations exposure. It's not clear if the box plots (figure 6) resume the data of all routes or only main route. The information given in figure 2 is not used to support quantitative discussion, only qualitative. I think it sould be interesting to show a figure with an example  with a typical profile  of UFP concentrations (10s average ) and traffic flow along the route. The box plots of each pair of SES could show the data divided into two parts: data registered in the segment of the main road and data of the segment of the quieter road. In figure 6 the legend in yy' axes is confused. Taking account the equiment range the logarithimic scale used is base e (ln ) not base 10 (log).

Like is given for traffic flow (figure 5) a figure with a bar plot concentrations for the 7/8 days of the five pairs  of routes and backgrounds needs to be included in order to be possible a comparison with the following figures. The method applied  to assess the traffic emission rates is row and the discussion is inconclusive. The road with must be added in order to assess the effect on the emission dilution. This parameter could be also included in the data normalisation.

I suggest to joint results and discussion in the same chapter, because in the results the authors included some comments that start the discussion of themself. 

Author Response

We appreciate the careful review and constructive suggestions. It is our belief that the manuscript is substantially improved after making the suggested edits. Please find below a point-by-point response to your concerns. We hope that you find our responses satisfactory and that the manuscript is now acceptable for publication. All line numbers refer to the revised manuscript file with highlighted changes.

Introduction - The introduction is too long with some irrrelevant redundancies. Paragraphs after line 67 can be cut in half.

Thank you for your suggestion. We eliminated some lines of the paragraph. Please check lines 69-77. It now reads: “The emission rate of vehicles can also vary significantly due to differences in a ve-hicle's engine technology, a vehicle’s age, mileage, vehicle model, level of maintenance, and driving behaviour [17]. Of these, the age of a vehicle is one of the most important factors, as older vehicles generally emit more pollutants due to accumulated mileage and their older engine technology (emission standards) [17-19]. Specifically, the higher accumulation of kilometres travelled results in deterioration of the engine and degra-dation of emission control equipment in combustion-based vehicles [19]. The gradual improvement in the emission standards of vehicles has meant that newer vehicles produce lower levels of emissions compared with their predecessors [17, 18]”.

Material and Methods - The authors give a detail description related to the traffic design assessment , however they presente a row description in relation to the UFP measurements. For exemple they never mention the distance from the ground to the  the UFP equipment inlet and how the equipement is transport along the road walk, as well the TSI particle model used TSI-P Track 8525) and operation cautions during transport in order to avoid interferences. The mode of log data acquisition depend the objectivs of measuement - source detection (every second), sample mode is recomended at least every 10 seconds (average measurement). 

Good point. Please see Lines 171-173. We added “ The device was set to collect data every second (1 s). During the data collection, the device was held at waist height to approximate the height of the breathing zone of a child”.

Data Analysis - the methodology described for the processing of UFP data is confusing and difficult to establish parallelism with the data presented in the Results chapter. They apply assumptions not tested and evaluated in relation to the observed concentrations ​​(example - background concentrations - sensitivity analisys). 

Thank you for the suggestion. Due to the lack of funding, we could not test the assumption. For example for testing the background air pollution, another operator was needed to measure background UFP at two different places. Please check the lines 224-234 for further explanation:

“For each day of data collection, the background air pollution during the time of data collection was estimated. The background PNC was measured immediately before and after data collection for five minutes (300 seconds) on a quiet road or in a park (located at least 200 meters away from any heavy traffic) close to the beginning of the first route and near the end of the second route. Given that background PNC is a function of time, a linear fit regression model () was used to estimate the background PNC during the period of data collection, assuming a linear trend over time between the start and end of data collection. The 600 data points (2×300 seconds background PNC) were used to determine the regression parameters (a and b in the equation above). The equation was then used to estimate the background PNC for every second of data collection”.

Results - a weak treatment of the UFP concentrations exposure. It's not clear if the box plots (figure 6) resume the data of all routes or only main route. The information given in figure 2 is not used to support quantitative discussion, only qualitative. I think it sould be interesting to show a figure with an example  with a typical profile  of UFP concentrations (10s average ) and traffic flow along the route. The box plots of each pair of SES could show the data divided into two parts: data registered in the segment of the main road and data of the segment of the quieter road. In figure 6 the legend in yy' axes is confused. Taking account the equiment range the logarithimic scale used is base e (ln ) not base 10 (log).

We fixed the scale of the figures. Please check Figure 7.

Good point. Thank you for your suggestion. A time series of UFP concentration for one of the routes is presented in Figure 11. 

Reviewer 3 Report

In general, the work is interesting, and represents some advancement over the actual state-of-the-art, the authors have collected a large database of measurement data. The issue has been widely developed, PM1 particle number concentrations along the children’s routes to schools, videos of the vehicles driving along these routs and data from GPS based tracking system were collected. The authors studied 5 pairs of schools (each pair consisted of a low and a high socioeconomic status). The main aim of the research was to determine whether the socio-economic status of schools influences the exposure of children attending particular schools to traffic-related pollution (PM1 number concentrations) while walking to school. The analysis took into account the age of cars, car flows and two normalized values. In fact, no significant differences were found  in children's exposure to PM1 en route to schools of different socioeconomic status.  The importance of these studies is local, while their organization  may be of interest to a wider group of readers. Therefore, I believe that the emphasis should be placed on the methodological part. I paid attention to a number of methodical ambiguities that require clarification before the manuscript would be suitable for publication.  Below, there are main  questions in order to make the manuscript clearer and more concise.

Abstract

  1. 22 - Please finish the sentence: “However neither ......
  2. 23-24 - These are not research results, but suggestions resulting from the analyzes

Materials and methods

I believe that it would be more precise to use the term PM1 instead of UFP, the latter are associated with particles with diameters below 100 nm. I do not deny that in technical studies there is a term for PM1 – UFP.

Fig. 1 - there is no pair  1.

  1. 137 - Please explain: How to understand: “ similar background levels of pollution”. After all, the variation in the background concentrations of PM1 was studied. Then, it was determined, so how was it assumed?
  2. 152-168 - For the clarity of the description and subsequent suggestions, I would compile a description of the roads characteristics (intersection, sidewalks, construction sites, etc.) in the form of a table (e.g. in Appendix)

l.172 - Please complete the device model

  1. 174 and further - Please note in the text for clarification: not the concentrations but the particle number concentrations were tested (particles cm-3).
  2. 2. – should be: Instrumentation

p.2. 3. – should  be: Data collection

In this section, please indicate the number of measurements and their organization.

  1. 223 227 - Please show on one example how the background value is determined. The current explanation is unclear. “ The background concentration was measured immediately before and after data collection for five minutes on a quiet road or in a park close to the beginning of the first route and near the end of the second route. Given  that background UFP is a function of time, a linear fit regression model (background(t)=a×time+b) was used to estimate the background concentration during the time of data collection, assuming a linear trend” - .- Unclear, was the background value set for each tour at the start and and at school of the second tour? Why is a background concentration a function of time? Ideally, the upwind distant site would serve as background monitoring sites. Adjusting concentration measurements for background is essential to reveal gradients of traffic related emissions. Please add, what averaging was taken for the background value.

L . 245 “ In this equation, ‘exposure’ is the average UFP concentration along the major road  (excluding the high exposures from traffic lights and constructions activities). “ Please explain it, how it was calculated.

Results

In section 3.1 please provide an overview of the results, concentration statistics, vehicles, etc. The current information has nothing to do with statistics.

The meteorological data is actually not used for analysis. If the authors want to prove their variability during the measurements, please include only the measurement periods in Fig. 2, e.g. in the form of a bar graph.

Fig. 2 -  no designation of the temperature axis

Why are the pairs numbers   shown in Figs 3-5, 7-8 in a different order? This makes interpretation difficult.

A paired t-test  was related to the mean values from each neighborhood of the school. Please indicate this, e.g. l 315, 316, 347, etc.

  1. 328 – „On the basis of a t-test, no significant difference was found in the mean exposure between low decile and high decile routes (t = 0.03, p = 0.98)” What data does this apply to, because in box- plots charts it appears that on some days they did not really differ, but on other days they most likely differed.

Fig. 6 - please check the scale on the y axis, too high concentration values beyond the measurement capabilities of the equipment. It would be clearer to present the  y axis: number concentrations (cm-3) and using a logarithmic scale.

Please add the explanation of Fig. 6 (the box-plot data).

Fig. 7 and 8 : units? Is it not clearer to express: (particles cm-3 / vehicles min-1)?

Fig. 9 should be described in more detail on what basis it was made. Please, for at least one route, show specific data in the charts (or in Appendix). The interpretation is not confirmed by the presented results.

Overall, it would be good to present  raw results for a single route with the indication of characteristic points. How were the spikes identified in time series? (1 second increase, or averaged values?). How was the cause determined for the 1 second increase?

In the Discussion part, the authors should refer more to the results obtained.  

Author Response

We appreciate the careful review and constructive suggestions. It is our belief that the manuscript is substantially improved after making the suggested edits. Please find below a point-by-point response to your concerns. We hope that you find our responses satisfactory and that the manuscript is now acceptable for publication. All line numbers refer to the revised manuscript file with highlighted changes.

Abstract

22 - Please finish the sentence: “However neither ......

We checked it to fix grammatical mistakes. Please see lines 22-23:

“However, neither measure of vehicle age was associated with a significant difference in exposures to PNC for pedestrians”.

23-24 - These are not research results, but suggestions resulting from the analyses

Thank you for your suggestion. We expected the routes characteristics had an important role in exposure to UFP. Please see lines 23-24:

“Route features, including narrow footpaths and a higher density of traffic lights, may be associated with higher levels of PNC exposure”.

Materials and methods

I believe that it would be more precise to use the term PM1 instead of UFP, the latter are associated with particles with diameters below 100 nm. I do not deny that in technical studies there is a term for PM1 – UFP.

Thank you for your suggestion. We used PNC instead of UFP as it is used by Rivas et al. (2017), too.

Rivas, I., Kumar, P., & Hagen-Zanker, A. (2017). Exposure to air pollutants during commuting in London: Are there inequalities among different socio-economic groups? Environ Int, 101, 143-157. 10.1016/j.envint.2017.01.019

Fig. 1 - there is no pair 1.

Good point. Fig.1 is changed to include pair 1, too.

137 - Please explain: How to understand: “ similar background levels of pollution”. After all, the variation in the background concentrations of PM1 was studied. Then, it was determined, so how was it assumed?

Thank you for the suggestion. Due to the lack of funding, we could not test the assumption. For example for testing the background air pollution, another operator was needed to measure background PNC at two different places.

152-168 - For the clarity of the description and subsequent suggestions, I would compile a description of the roads characteristics (intersection, sidewalks, construction sites, etc.) in the form of a table (e.g. in Appendix)

Characteristics of the routes are now mentioned in table 2. Please see table 2.

l.172 - Please complete the device model

Please see line 168. We changed it from “P-Trak” to “P-Trak particle counter 8525”.

174 and further - Please note in the text for clarification: not the concentrations but the particle number concentrations were tested (particles cm-3).

It is corrected as suggested. We changed it from “UFP” to “PNC” all over the manuscript.

  1. – should be: Instrumentation

It is corrected as suggested. Please see the line 165. We changed the subheading from “Data collection and instrumentation” to “Instrumentation”.

p.2. 3. – should  be: Data collection

In this section, please indicate the number of measurements and their organization.

It is corrected as suggested. Please see line 188. We changed the subheading from “time” to “data collection”.

We added some explanation about the measurements. Please see lines 196-199:

“The researcher traversed five pairs of routes over 39 days (a total of seven days for Pair 2 and eight days each for the remainder). Over the data collection period, a total of 83 kilometres was travelled”.

223 227 - Please show on one example how the background value is determined. The current explanation is unclear. “ The background concentration was measured immediately before and after data collection for five minutes on a quiet road or in a park close to the beginning of the first route and near the end of the second route. Given  that background UFP is a function of time, a linear fit regression model (background(time)=a×time+b) was used to estimate the background concentration during the time of data collection, assuming a linear trend” - .- Unclear, was the background value set for each tour at the start and and at school of the second tour? Why is a background concentration a function of time? Ideally, the upwind distant site would serve as background monitoring sites. Adjusting concentration measurements for background is essential to reveal gradients of traffic related emissions. Please add, what averaging was taken for the background value.

Thank you for your suggestion. Due to funding issues, only one person (the researcher) collected data. Therefore, the background air pollution could only be collected before starting data collection for the first route and after data collection for the second route.

A quiet street or a park 300 metres away from traffic was chosen to collect background PNC. Please see line 224 to 234:

“For each day of data collection, the background air pollution during the time of data collection was estimated. The background PNC was measured immediately before and after data collection for five minutes (300 seconds) on a quiet road or in a park (located at least 200 meters away from any heavy traffic) close to the beginning of the first route and near the end of the second route. Given that background PNC is a function of time, a linear fit regression model () was used to estimate the background PNC during the period of data collection, assuming a linear trend over time between the start and end of data collection. The 600 data points (2×300 seconds background PNC) were used to determine the regression parameters (a and b in the equation above). The equation was then used to estimate the background PNC for every second of data collection”.

L . 245 “ In this equation, ‘exposure’ is the average UFP concentration along the major road  (excluding the high exposures from traffic lights and constructions activities). “ Please explain it, how it was calculated.

We added some lines to cover it. Please see 252-254:

“(excluding the high exposures from traffic lights and constructions activities, and the high exposures from these places were removed from the excel file using the GPS coordination, the time of data collection and the videos)”

Results

In section 3.1 please provide an overview of the results, concentration statistics, vehicles, etc. The current information has nothing to do with statistics.

We eliminated this section as explaining about statistics of the concentration or the traffic flow could be redundant.

The meteorological data is actually not used for analysis. If the authors want to prove their variability during the measurements, please include only the measurement periods in Fig. 2, e.g. in the form of a bar graph.

We added two barographs to show the variability of temperature and wind speed. Please see Figure 3.

Fig. 2 -  no designation of the temperature axis

Good point. The Fig.2 is corrected as suggested. 

Why are the pairs numbers   shown in Figs 3-5, 7-8 in a different order? This makes interpretation difficult.

The bars are sorted ascending based on the values for high decile routes. 

A paired t-test  was related to the mean values from each neighborhood of the school. Please indicate this, e.g. l 315, 316, 347, etc.

In all cases, the paired t-test is related to the mean values from each school’s neighbourhood, unless it is mentioned.

328 – „On the basis of a t-test, no significant difference was found in the mean exposure between low decile and high decile routes (t = 0.03, = 0.98)” What data does this apply to, because in box- plots charts it appears that on some days they did not really differ, but on other days they most likely differed.

Good point. We added a bar plot to make it clear. Please see Figure 8.

Fig. 6 - please check the scale on the y axis, too high concentration values beyond the measurement capabilities of the equipment. It would be clearer to present the  y axis: number concentrations (cm-3) and using a logarithmic scale.

Thank you for suggestion. We fixed it. Please check Figure 7.

Please add the explanation of Fig. 6 (the box-plot data).

We added some explanation. Please check the lines 310-314:

“Figure 7 presents a boxplot of the log-transformed PNC exposure for the whole of the routes. The results suggest that there is significant variability in measured PNC from one day to another for a given route. The lowest and highest daily average PNC were 1,900 ± 3,600 pt/cc (the high decile route of Pair 1 on Day 1) and 57,700 ± 24,100 pt/cc (the low decile route of Pair 4 on Day 1)".

 Fig. 7 and 8 : units? Is it not clearer to express: (particles cm-3 / vehicles min-1)?

We fixed the y-label as suggested. Please see Figure 9 and Figure 10.

 Fig. 9 should be described in more detail on what basis it was made. Please, for at least one route, show specific data in the charts (or in Appendix). The interpretation is not confirmed by the presented results. Overall, it would be good to present  raw results for a single route with the indication of characteristic points. How were the spikes identified in time series? (1 second increase, or averaged values?). How was the cause determined for the 1 second increase?

Thank you. We added the bar plot for the number of spikes for each route. An example of a time series and the reason for spikes is also presented. Please check Figure 11 and Figure 13.

 In the Discussion part, the authors should refer more to the results obtained.

We tried to refer to the results in Discussion as much as possible.

Reviewer 4 Report

The current manuscript describes a study conducted to access the potential levels of traffic-generated ultrafine particulate matter (UFP)-exposure to children walking to school in different regions of Auckland with differing socioeconomic status.  The results of the current study are well-presented; however, there are a few minor comments that need to be further addressed by the authors.

Comments:

  1. Please indicate statistical relationships on graphs with use of an asterisk (*), or other defined symbol.
  2. Please check references.  There are a few locations throughout the manuscript that the following replaces the references cited:  "Error! Reference source not found."
  3. The abstract (and perhaps the title) should mention the location of the study.  This is important as weather patterns, traffic-patterns, types of fuel used in vehicles, make/age of vehicles in a regions, etc. are all important to the study outcomes.  Thus, similar studies may produce very different results across different regions.

Author Response

We appreciate the careful review and constructive suggestions. It is our belief that the manuscript is substantially improved after making the suggested edits. Please find below a point-by-point response to your concerns. We hope that you find our responses satisfactory and that the manuscript is now acceptable for publication. All line numbers refer to the revised manuscript file with highlighted changes.

  1. Please indicate statistical relationships on graphs with use of an asterisk (*), or other defined symbol.

The significant difference was observed only in age of vehicles. Therefore, we thought it may be a better option to mention the significant difference in the text.

  1. Please check references.  There are a few locations throughout the manuscript that the following replaces the references cited:  "Error! Reference source not found."

Thank you for the suggestion. We fixed the references as it is suggested.

  1. The abstract (and perhaps the title) should mention the location of the study.  This is important as weather patterns, traffic-patterns, types of fuel used in vehicles, make/age of vehicles in a regions, etc. are all important to the study outcomes.  Thus, similar studies may produce very different results across different regions.

Good point. The title is changed as suggested: “Socioeconomic status and route characteristics in relation to children’s exposure to air pollution from road traffic while walking to school in Auckland, New Zealand”.

We added the location of study in the abstract and title. Please refer to lines 14-17: “In this study, we investigated the effects of differences in vehicle fleet and transport infrastructure on children's exposure to traffic-related air pollution during their morning walk to school in Auckland, New Zealand”.

Round 2

Reviewer 3 Report

I accept the authors' explanations and believe that the clarity of the text has improved.